# Nicotine in Combination with SARS-CoV-2 Affects Cells Viability, Inflammatory Response and Ultrastructural Integrity

**DOI:** 10.3390/ijms23169488

**Published:** 2022-08-22

**Authors:** Luigi Sansone, Antonio de Iure, Mario Cristina, Manuel Belli, Laura Vitiello, Federica Marcolongo, Alfredo Rosellini, Lisa Macera, Pietro Giorgio Spezia, Carlo Tomino, Stefano Bonassi, Matteo A. Russo, Fabrizio Maggi, Patrizia Russo

**Affiliations:** 1MEBIC Consortium, San Raffaele University, 00166 Rome, Italy; 2Cellular and Molecular Pathology, IRCCS San Raffaele Roma, Via di Val Cannuta 247, 00166 Rome, Italy; 3Experimental Neurophysiology, IRCCS San Raffaele Roma, Via di Val Cannuta 247, 00166 Rome, Italy; 4Department of Human Sciences and Quality, Life Promotion San Raffaele University, Via di Val Cannuta 247, 00166 Rome, Italy; 5Department of Molecular Medicine, University La Sapienza, Viale del Policlinico 155, 00161 Rome, Italy; 6Department of Clinical, Internal, Anesthesiologist and Cardiovascular Sciences, La Sapienza University, Viale del Policlinico 155, 00161 Rome, Italy; 7Laboratory of Flow Cytometry, IRCCS San Raffaele Roma, Via di Val Cannuta 247, 00166 Rome, Italy; 8Clinical and Molecular Epidemiology, IRCCS San Raffaele Roma, Via di Val Cannuta 247, 00166 Rome, Italy; 9Virology Division, Pisa University Hospital, Via Paradisa 2, 56127 Pisa, Italy; 10Scientific Direction, IRCSS San Raffaele Roma, Via di Val Cannuta 247, 00166 Rome, Italy; 11Laboratory of Virology, IRCCS National Institutes for Infectious Diseases “Lazzaro Spallanzani” INMI, Via Portuense 292, 00149 Rome, Italy

**Keywords:** cytokines, human lung adenocarcinoma A549 cells, necroptosis, nicotine, poly(I:C), pyroptosis, proinflammatory cytokines, SARS-CoV-2

## Abstract

The aims of our study are to: (i) investigate the ability of nicotine to modulate the expression level of inflammatory cytokines in A549 cells infected with SARS-CoV-2; (ii) elucidate the ultrastructural features caused by the combination nicotine+SARS-CoV-2; and (iii) demonstrate the mechanism of action. In this study, A549 cells pretreated with nicotine were either exposed to LPS or poly(I:C), or infected with SARS-CoV-2. Treated and untreated cells were analyzed for cytokine production, cytotoxicity, and ultrastructural modifications. Vero E6 cells were used as a positive reference. Cells pretreated with nicotine showed a decrease of IL6 and TNFα in A549 cells induced by LPS or poly(I:C). In contrast, cells exposed to SARS-CoV-2 showed a high increase of IL6, IL8, IL10 and TNFα, high cytopathic effects that were dose- and time-dependent, and profound ultrastructural modifications. These modifications were characterized by membrane ruptures and fragmentation, the swelling of cytosol and mitochondria, the release of cytoplasmic content in extracellular spaces (including osmiophilic granules), the fragmentation of endoplasmic reticulum, and chromatin disorganization. Nicotine increased SARS-CoV-2 cytopathic effects, elevating the levels of inflammatory cytokines, and inducing severe cellular damage, with features resembling pyroptosis and necroptosis. The protective role of nicotine in COVID-19 is definitively ruled out.

## 1. Introduction

The global pandemic of the COronaVIrus Disease 2019 (COVID-19) is the result of infection with the severe acute respiratory syndrome coronavirus-2 (SARS-CoV-2). It was first isolated and identified in patients in Wuhan City, Hubei Province, China on December 2019 [1]. SARS-CoV-2, such as SARS-CoV and MERS-CoV-2, is an enveloped, positive-sense, single-stranded RNA virus of the genus β-coronavirus, and it shares 79% nucleotide sequence identity to SARS-CoV [2]. SARS-CoV-2 relies on its obligate receptor, angiotensin-converting enzyme 2 (ACE2), to enter cells mediated by the S (Spike) glycoprotein. Indeed, ACE2 is the only confirmed SARS-CoV-2 entry receptor [3,4]. ACE2 works in concert with furin-like proteases, cathepsin L, and the host cell transmembrane protease serine 2 (TMPRSS2) (recently reviewed by Jackson et al. [4]). TMPRSS2 is a type II transmembrane serine protease, which cleaves the spike protein of SARS-CoV-2, contributing to viral-host membrane fusion and subsequent infection. ACE2 relies on the RAS molecular system; it is a crucial counter-regulatory enzyme to ACE by breaking down angiotensin II, which is involved in blood pressure regulation and electrolyte homeostasis [5]. During SARS-CoV-2 infection, patients experience systemic symptoms of varying severity that range from few or no symptoms to severe pneumonia, which can further progress to acute respiratory distress syndrome (ARDS) and death. These outcomes are associated with an aggressive inflammatory response and the release of large amounts of pro-inflammatory cytokines such as interleukins 6,8,10, or TNF-α; this release is called a cytokine storm [6]. NLRP3 inflammasome is activated in response to SARS-CoV-2 infection, and is active in COVID-19 patients as it can be found in PBMCs and lung tissues of postmortem patients upon autopsy [6]. Deaths in patients with severe COVID-19 are strictly related to elevated levels of circulating pro-inflammatory cytokines [7,8]. Studies have suggested a direct correlation between cytokine storms and lung injury, multi-organ failure, and poor prognosis [9,10]. Recently, The Lancet Respiratory Medicine published a shocking editorial stating that “the consequences of the COVID-19 pandemic have worsened an already unacceptable global burden of disease for respiratory cancer” [11], introducing the concept of a “dangerous liaison” between COVID-19, smoking, and cancer. This statement is based on the systematic analysis made by the Global Burden of Disease Study in 2019, who reported that the number of cases and deaths from cancer of the trachea, bronchi, lungs, and larynx has increased globally over the past decade. According to the editorial [11], the consequences of the COVID-19 pandemic have worsened an already unacceptable global burden of respiratory cancer. Moreover, during the 18th WCTOH (World Conference on Tobacco or Health) held in Dublin, Ireland in March 2022, Silvano Gallus, Janice Leung, and Catherine Egbe, invited as expert speakers, stated that “There is an urgency to understand the relationship between COVID-19 and tobacco use” [12]. Thus, while the COVID-19 pandemic could have represented an opportunity to reduce smoking rates globally, the number of smokers has actually risen dramatically both in the USA and Europe during the lock-down period [13]. Although, multifactorial reasons may explain this increase, the debate over the possible protective effect of tobacco smoke (and of nicotine in particular) on disease onset/severity definitely did not encourage individuals to quit smoking. During different phases of the pandemic, some reports suggested that current smokers are less likely to be infected by SARS-CoV-2, contributing to the beliefs of tobacco smokers and e-cigarette users that their habit did not increase the risk of severe COVID-19 [14]. Most smokers (84%) appear reluctant or indecisive in taking the vaccine, raising concerns about COVID-19 vaccination [15]. In this scenario, tobacco companies were quick to capitalize on the opportunities offered by the so-called protective “nicotinic hypothesis” [16]. This theory relies on the observation that SARS-CoV-2 could interact with nicotinic acetylcholine receptors (nAChR) triggering nicotinic cholinergic anti-inflammatory system dysregulation [16]. Although larger and more accurate prospective population cohorts showed that smoking increases the risk, the severity, and the mortality for COVID-19 [17], three clinical trials based on nicotine therapy [18,19,20] are ongoing. The lack of published results from these trials is further remarked on as a positive effect of tobacco products (i.e., nicotine) on COVID-19, though this is far from being demonstrated. In our previous experiments, we suggested that nicotine, at concentrations mimicking those present in the human plasma after smoking one cigarette (i.e., 0.1 µM) [21] in A549 cells: (i) enhances the expression levels of α7-nAChR; (ii) is not cytotoxic; (iii) up-regulates the angiotensin-converting enzyme-2 (ACE2) expression at the mRNA/protein level; (iv) increases SARS-CoV-2 replication and transcription; and (v) increases the SARS-CoV-2 cytopathic effect. Despite the growing evidence describing a synergistic effect of nicotine on SARS-CoV-2 spreading, the complete mechanism of action of this effect has not yet been described. Thus, we explored, at the ultrastructural level, the effects induced by nicotine on SARS-CoV-2 spreading in A549 cells. Moreover, we investigated the ability of nicotine to modulate the expression level of TNFα, IL6, IL8, and IL10 in A549 cells infected with SARS-CoV-2. A comparative analysis describing the replication features of SARS-CoV-2 in A549 and Vero E6 cells, one of the most commonly used cell lines for studying this virus [22], is reported. In this work, for the first time, we advance the hypothesis that PANoptosis is involved in the synergistic action of nicotine+SARS-CoV-2, and we contribute decisive evidence demonstrating the dangerous effect of nicotine when associated with SARS-CoV-2.

## 2. Results

### 2.1. Evaluation of Cytokine Release in Control, Nicotine-Treated, and SARS-CoV-2 Infected Cells

LPS or poly(I:C) stimulation for 12 h alone showed higher expression of TNFα and IL6 proteins in A549 cells when compared to untreated cells (Figure 1A). Nicotine pre-treatment produced an inhibitory effect on the expression of TNFα proteins with a 38.8% reduction in poly(I:C) treated cells. The expression of the IL6 protein was also reduced, both in cells treated with LPS [−46.3%] and with poly(I:C) [−36.3%] (Figure 1A). Treatment with LPS (10 μg/mL for 12 h) or with poly(I:C) (20 μg/mL alone for 12 h) did not induce cytotoxicity (data not shown), which was the same as in the presence of 0.1 μM nicotine alone.

Interestingly, the presence of the competitive antagonist BTX (bungarotoxin) counteracted the effect induced by nicotine on poly(I:C)-induced TNFα (Figure 1A).

When A549 cells were pre-treated for 24 h with 0.1 µM nicotine and then infected with 2 TCID50 of SARS-CoV-2 for 24 h, levels of IL6, IL8, IL10, and TNFα were evaluated on supernatants. Vero E6 cells were used as a positive control. When Vero E6 cells were exposed to 20 TCID50 of SARS-CoV-2 for 24 h, the expression levels of all cytokines increased dramatically as compared to untreated cells (+546.8% for IL6, +193.9% for IL8, +220.0% for IL10, and +211.4% for TNFα, Figure 1B).

In A549 cells, pretreatment with nicotine increased the levels of SARS-CoV-2 induced cytokines dramatically (+101.9% for IL6, +17.9% for IL8, +178.1% for IL10, and +130.4% for TNFα, Figure 1B).

As shown in Figure 1B, the levels of cytokines in A549 cells were similar to those induced in control Vero E6 cells similarly pretreated with nicotine and exposed to SARS-CoV-2.

### 2.2. Cell Cytotoxicity

The cytotoxic effects of different treatments were measured both in Vero E6 and in A549 cells. Nicotine pretreatment alone did not induce cytotoxicity (Figure 1C,D). When nicotine-pretreated cells were incubated with SARS-CoV-2, the percentage of cell depletion dramatically increased (Figure 1C,D). This effect is very high in Vero E6 (Figure 1C).

Consistent with our previous data [21], A549 cells were less responsive to the virus (Figure 1D); exposure to 20 TCID50 for 24 h induced a very moderate cytopathic effect (30.3 ± 1.8%, Figure 1D) and a very low effect at 10 TCID50 (8.7 ± 0.9%, Figure 1D). When A549 cells were pre-exposed to nicotine, the effect increased dramatically in a dose-/time-dependent manner (Figure 1D). The cytopathic effect was similar to that observed in Vero E6 cells (Figure 1C).

### 2.3. Ultrastructural Changes at TEM

TEM was used to analyze the ultrastructural changes induced by nicotine alone, by SARS-CoV-2 (20 TCID50 for 24 h) alone, and by both treatments. The replicative features of SARS-CoV-2 in A549 and Vero E6 cells were comparatively analyzed.

Control Vero E6 cells were well preserved (Figure 2A).

Usually, they were grown as a monolayer, occasionally forming overlapping structures. Basal and apical poles were easily recognizable (as indicated by white bar with arrowheads, Figure 2A(a)). All nuclear and cytoplasmic components displayed normal ultrastructure, including the nuclear membrane, mitochondria, endoplasmic reticulum, and plasma membrane; the latter at the apical pole released vesicles of different sizes (from exosomes to exophers). Occasionally, lipid granules (immature surfactant) were present (Figure 2A(b), arrows).

SARS-CoV-2 infected Vero E6 cells presented a variable number of viral particles, depending on different cells (Figure 2A(b–d)). Virus particles were localized in single membrane-bound vacuoles (Figure 2A(b)) or the extracellular spaces, as single particles or in groups of different sizes (Figure 2A(c)). The morphology of virus particles varied in relation to the phase of maturation and release from the infected cell. The heads of spike proteins (corona) were more clearly evident in assembled particles. Subcellular components showed various degrees of ultrastructural disorganization, due to the cytopathogenic effects of the virus infection. Occasionally, several autophagosomes, and some mitochondrial alterations, increased vesicle release at the apical pole, and necrotic cells were seen (Appendix A).

The ultrastructure of untreated A549 cells showed two distinct cell subpopulations (Figure 2B). The first grew mostly in well-ordered monolayers as Type 1 pneumocytes (Figure 2B(a)). In the second, the majority of cells were similar to Type 2 pneumocytes (Figure 2B(b)), showing basal and apical poles, and osmiophilic large lipid granules (likely containing unsaturated lipid precursors of surfactant) mostly associated with the Golgi and endoplasmic reticulum. Mitochondria were well preserved in orthodox form and were abundant, especially in more differentiated cells. Osmiophilic lipid bodies were often strictly associated at the apical pole of the plasma membrane for secretion, or released in the extracellular space, as contained in a membrane-bound vesicle (Figure 2B(b)).

A549 cells underwent three major changes after nicotine treatment, although remained well-preserved and organized: 1. The cell area appeared substantially increased with large nuclei and cytoplasm, suggesting a hypertrophic change; 2. The cytoplasm contained a large number of osmiophilic lipid bodies as compared to the untreated cells. 3. Cells without osmiophilic granules (similar to Type 1 pneumocytes) were quite rare/decreased. Three types of granules were observed: 1. Very dense and homogeneous; 2. Decreased density and homogeneity; 3. Vacuoles containing myelin-like osmiophilic membranes (final differentiation) (Figure 3a–d).

In A549 cells infected with SARS-CoV-2 alone, no definite and mature virus particles were detected. However, a large number of ultrastructural cytopathic virus-associated features were observed. These included: (a) an increased number of multivesicular bodies thought to be elements for the assembly of virus particles; (b) some necrotic disorganized cells with membrane fragmentation or large-pore nuclear changes, especially chromatin and nuclear membrane disorganization; and (c) the release of plasmamembrane vesicles of various sizes (Figure 4A).

In A549 cells treated with SARS-CoV-2 and nicotine, the majority of cells were necrotic or in various degrees of disorganization. The following prominent ultrastructural features were observed: (a) membrane rupture/fragmentation with a number of large pores (Figure 4B + insert); (b) swollen cytosol and mitochondria, fragmentation of endoplasmic reticulum; (c) the presence of numerous autophagosomes containing membrane debris, myelin-like bodies and various cytosolic content (Figure 4B); (d) the release of cytosolic content in extracellular space, including osmiophilic granules; (e) chromatin and nuclear membrane disorganization; and (f) in a few best-preserved cells, plasma membranes were involved in vesicle formation with single/rare viral particles (Figure 4B).

Although few viral particles have been found, the quantitative real-time PCR detected SARS-CoV-2 specific E, RdRp, and N gene in the supernatants of these cells, as shown in our previous work [21].

## 3. Discussion

We present new experimental findings supporting our original hypothesis that nicotine enhances COVID-19 severity [23]. Thus, nicotine, at concentrations mimicking the human exposure in smokers (i.e., concentration and exposure time) in A549 cells exposed to SARS-CoV-2 produces the substantial induction of IL6, IL8, IL10 and TNFα, increased cytotoxicity; and complex ultrastructural modifications. We have previously shown that nicotine enhances the expression of α7-nAChR and ACE2 through MAPK/ERK activation, both on human bronchial epithelial cells (HBEpC) [23] and on A549 cells [24]. All effects were strictly dependent on α7-nAChR, since when α7 was silenced (si-mRNA-α7-HBEpC) or blocked by α-bungarotoxin, no ACE2 upregulation was observed [23]. In A549 cells, nicotine increases SARS-CoV-2 replication, transcription of viral proteins, and cytopathic effects [21]. We assumed that the increased entry and replication of SARS-CoV-2 may be caused by the effects of α7-nAChR-mediated signaling on ACE2 expression, as formerly hypothesized by Leung et al. [25,26] and ourselves [21,23,24]. A549 shows features of Type 2 (AT2, progenitor cells for T1) cells of the alveolar epithelium, including osmiophilic dense/lamellar bodies, identified as granules of immature/maturing surfactant [27]. A second, less abundant cell population is characterized by the absence of electron-dense bodies, an increased nucleus/cytoplasm ratio and scarce organelles in the cytoplasm. When the lung is injured, AT2 cells divide and cover the injured area, allowing the gradual reemergence of AT1 cells [28]. Although the A549 cell line is commonly used as an AT2 cell model, these cells are incompatible with SARS-CoV-2 infection (less than 1% of infection) [29]. A possible mechanism of A549 resistance is the lower amount of ACE2, of transmembrane serine protease 2 (TMPRSS2), and furin as compared to Vero E6. Accordingly, experiments with SARS-CoV-2 were performed in human ACE-transfected A549 cells (A549ACE2+) to obtain overexpression of ACE2 mRNA/proteins and subsequent efficient virus infection. In our model, nicotine pretreatment allowed sensitization of A549 wild-type to SARS-CoV-2 infection as ACE2 mRNA are copiously induced through α7-nAChR/MAPK/ERK/phospho-p38/p53 pathway activation [21,23,24]. Under this condition, 20 TCID50 of SARS-CoV-2 was the optimal dose for investigating differences in virus replication and cytopathic effects [21]. Thus, the cytopathic effects induced by SARS-CoV-2 was dose (TCDI50)- and time-dependent, and included severe subcellular changes and massive cell death. Terminal cell death was associated with membrane rupture and fragmentation, swelling of cytosol and mitochondria, the release of cytoplasmic content in extracellular spaces (ECS) (including osmiophilic granules), the fragmentation of endoplasmic reticulum, and chromatin disorganization. The observed abundance of death disorganization was in agreement with the quantitative measurements of cytotoxicity. Moreover, after nicotine+SARS-CoV-2 treatment, cells released high levels of TNFα, IL6, IL8, and IL10, similarly to what is observed in COVID-19 patients with high disease severity and mortality [30]. On the other hand, unlike what we observed in the presence of SARS-CoV-2, pretreatment of A549 cells with nicotine, before stimulation with LPS [Toll-Like Receptor (TLR) 4 agonist), or poly(I:C) (TLR3 agonist), was able to reduce the expression levels of IL6 and TNFα. This effect seems specifically mediated by nicotine, since in the presence of BTX no decrease in poly(I:C)-induced cytokines was observed. We observed this effect also in human HBEpC (data not shown). Further, these data are in agreement with the inhibitory effect of nicotine on the HBE16 cell line stimulated by LPS through an α7-nAChR/MyD88/NF-ĸB pathway [31]. TLR4 and TLR3 were involved in the inflammation induced by LPS and poly(I:C), respectively [32], whereas the receptors involved by SARS-CoV-2 are principally TLR7, TLR8 [33], and especially the NLRP3 inflammasome [34,35]. The diverse pathways activated by these proinflammatory receptors could explain the different subcellular changes, the different type of cell deaths, and different cytokine production after stimulation by the virus alone or by LPS or poly(I:C), although their different steps may converge in the final common death pathways as shown. According to our results, the most likely mechanisms linking the over-release of cytokine to organ damage leading to cell death, are pyroptosis, apoptosis, or necroptosis. This is a hypothesis which has been only partially described in previous studies. Several innovative details have been provided by the present research, which adds decisive evidence on PANoptosis and to its sub-processes. Pyroptosis and necroptosis are inflammatory cell death processes characterized by lytic forms of death in which the release of cytokines such as IL1, IL6, IL8, and TNFα trigger inflammation and prompt an immune response, whereas apoptosis is classically considered to be immunologically mute [36,37]. Thus, depending upon the stimulus encountered (bacterial/viral infection, or chemical insults), cells can experience extensive crosstalk between pyroptosis, apoptosis, and necroptosis, leading to PANoptosis, a unique, physiologically relevant inflammatory programmed cell death pathway, activated by specific triggers and regulated by the PANoptosome complex. Recently, it was shown that among all cytokines induced by SARS-CoV-2 infection, TNFα and IFNγ played a prominent role in damaging vital organs by inducing inflammatory cell death (PANoptosis) [38]. In this work, we showed that nicotine is able to dramatically increase the levels of cytokines including TNF-α. The simultaneous presence of SARS-CoV-2 may induce pyroptosis and necroptosis, which supported the hypothesis that this is the most likely cellular and molecular mechanism pursued by these cytokines to generate the cytokine storm, resulting in inflammatory cell death. While it is well known that SARS-CoV-2 infection induces monocytes’ release of proinflammatory cytokines, and subsequently induce pyroptosis, it is less clear whether other cell types in lung tissue, such as alveolar epithelium, undergo apoptosis though this mechanism [39]. We showed that nicotine induces pyroptosis in human bronchial epithelial cells (A549 wild-type cells) infected by SARS-CoV-2. This hypothesis is supported by a number of references showing that SARS-CoV-2 (and many other viruses) [34] and nicotine—though at much higher concentrations [40]—activated NLRP3 inflammasome, and produced pyroptosis, necroptosis, apoptosis, and cytokine production, activation, and release. It is unclear if the effects of nicotine and the virus are independent or synergistic. Pyroptosis and necroptosis are proinflammatory forms of cell death, characterized by the formation of pores in the plasma membrane (Appendix A), although through a different post-NLRP3-activation pathway. In pyroptosis, the NLRP3/Caspase-1 pathway leads to Gasdermin D cleavage by caspase-1 to generate pore-forming peptides [41]. In necroptosis, it is the NLRP3/RIPK1/2 pathway that leads to the activation of MLKL (mixed lineage kinase domain-like pseudokinase) peptide which can polymerize, forming membrane pores [42].

## 4. Materials and Methods

### 4.1. Cells and Chemicals

The human lung adenocarcinoma cell line A549 (CCL-185 ATCC) and VERO C1008 [Vero 76, clone E6, Vero E6] (CRL-1586 ATCC) were purchased by ATCC and grown as monolayers following the ATCC’s protocols.

### 4.2. SARS-CoV-2 Culture

An inoculum was used a SARS-CoV-2 positive cell supernatant. The cell supernatant, obtained by infecting Vero E6 cells with a nasopharyngeal swab of a patient with SARS-CoV-2 infection, contained 2000 tissue culture infectious dose 50 (TCID50), as estimated by endpoint titration. Adherent A549 and Vero E6 cell lines, grown in appropriate temperature conditions, and RPMI 1640 medium supplemented with 10% heat-inactivated FBS and 1% penicillin/streptomycin, were exposed to the viral inoculum (0.5 mL) in the form of traditional culture when cell monolayers were less than two days old. After infection, the inoculum was removed, the monolayers rinsed three times with sterile phosphate-buffered saline (PBS), and 3 mL of the appropriate culture medium was added. All culture plates were incubated in a humidified 37 °C incubator in an atmosphere of 5% CO_2_. Cells were monitored daily for the development of cytopathic effects (CPE). After three days, 400 µL of cell supernatant was used for total nucleic acid extraction for SARS-CoV-2 RNA testing. Negative control samples were supernatants obtained from A549 cells not infected with SARS-CoV-2.

### 4.3. Cell Viability Assay

For determining the optimal experimental dose of nicotine, and for evaluating the proliferation of the cells under nicotine treatment, the viability of the pulmonary A549 cells was assessed. After cell suspension inoculation (100 µL/well) in a 96-well plate, the plate was pre-incubated in a humidified incubator (at 37 °C, 5% CO_2_). Then, 10 µL of the CCK-8 solution was added to each well of the plate. After incubation for 1–4 h, the absorbance was measured at 450 nm using a microplate reader. A calibration curve using the data obtained from the wells containing known numbers of viable cells was prepared. The viability was calculated as the percentage ratio between treated cells and control cells. Cytotoxicity was determined by Cell Counting Kit-8 (Catalog number: 96,992, Sigma-Aldrich, St. Louis, MO, USA). After dispensation of 100 µL of cell suspension (5000 cells/well) in a 96-well plate, the solution was pre-incubated for 24 h in a humidified incubator (at 37 °C, 5% CO_2_). Ten µL of various concentrations of toxicant were added to the culture media in the plate. After incubation of the plate for an appropriate length of time (6, 12, 24, 48 h), 10 µL of CCK-8 solution was added to each well of the plate, being careful not to introduce bubbles to the wells, since they interfere with the O.D. reading. After incubation of the plate for 1–4 h, the absorbance at 450 nm was measured by using a microplate reader.

### 4.4. Markers of Inflammation

IL6, IL8, IL10, and TNFα concentrations were measured using Human IL6 DuoSet ELISA [Catalog number: DY206-05; Range: 9.4–600 pg/mL], Human IL8/CXCL8 DuoSet ELISA [Catalog number: DY208; Assay Range: 31.2–2000 pg/mL], Human IL10 DuoSet ELISA [Catalog number: DY217B; Assay Range: 31.2–2000 pg/mL], and Human TNFα DuoSet ELISA [Catalog number: DY210; Assay Range: 15.6–1000 pg/mL], respectively. All were purchased by R&D Systems (Minneapolis, MN, USA) following the manufacturer’s protocols.

### 4.5. Transmission Electron Microscopy

Biological samples were collected and fixed overnight in 2% glutaral-dehyde with 1% tannic acid in 0.1 M sodium cacodylate, pH. Samples were rinsed 6 times in the sodium cacodylate buffer, and then incubated in 2% osmium tetroxide in the same buffer for 2 h at room temperature and processed following a standard schedule for embedding in EPON resin. Following polymerization overnight at 65 °C, 80 nm sections were cut on a Righter-Jung Ultra cut E Ultramicrotome (Leica Microsystems, Wetzlar, Germany) and picked up on copper grids to be analyzed in a TEM-1400 Plus.

### 4.6. Statistical Analysis

Data were managed and analyzed using GraphPad Prism 8.1 (GraphPad Software Inc., La Jolla, CA, USA). Mean values of independent samples were compared using the Student’s *t* test, or with one-way ANOVAs for groups with n > 2, using post hoc Bonferroni corrections to take into account the effect of multiple comparisons. Experiments were performed at least two times in triplicate. A *p*-value < 0.05 was considered statistically significant.

## 5. Conclusions

Taken together, these findings greatly weaken the hypothesis that the nicotine α7-nAChR-mediated cholinergic anti-inflammatory pathway would be able to attenuate inflammatory lung injury produced by SARS-CoV-2 infection by inhibiting cytokine release, as claimed by different authors [43]. The availability of a reliable mechanism that characterizes the negative effect of cigarette smoking on the severity and progression of COVID-19 may provide not only a solid scientific basis for understanding the dangerous relationship between COVID-19 and tobacco, but also an additional public health protection measure.

## Figures and Tables

**Figure 1 ijms-23-09488-f001:**
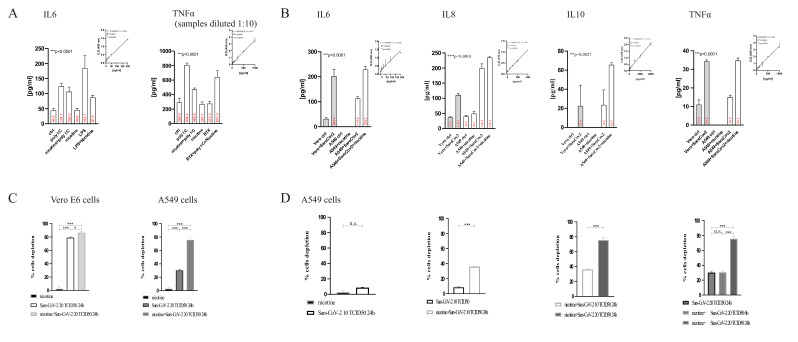
(**A**) Evaluation of IL6 and TNFα increase in A549 cells stimulated with LPS or poly(I:C). ELISA experiments; regression equation linearity, performed with Prism. Statistical significance was analyzed using one-way ANOVAs with multiple-comparison and post hoc tests with Bonferroni corrections. (**B**) Evaluation of IL6, IL8, IL10, or TNFα increase in A549 cells exposed to SARS-CoV-2. When histogram is not present it means that the values were below the sensitivity threshold of the test. See Materials and Methods for sensitivity thresholds for IL6, IL8, IL10, and TNFα, respectively. ELISA experiments; regression equation linearity, performed with Prism. Statistical significance was analyzed with one-way ANOVAs with multiple-comparisons and post hoc tests with Bonferroni corrections. (**C**,**D**) Evaluation of cell cytotoxicity in Vero E6 (**C**), and A549 cells (**D**). *** *p* < 0.001, ** *p* < 0.01, * *p* < 0.1; n.s. not significant *p* > 0.05 it is the correction.

**Figure 2 ijms-23-09488-f002:**
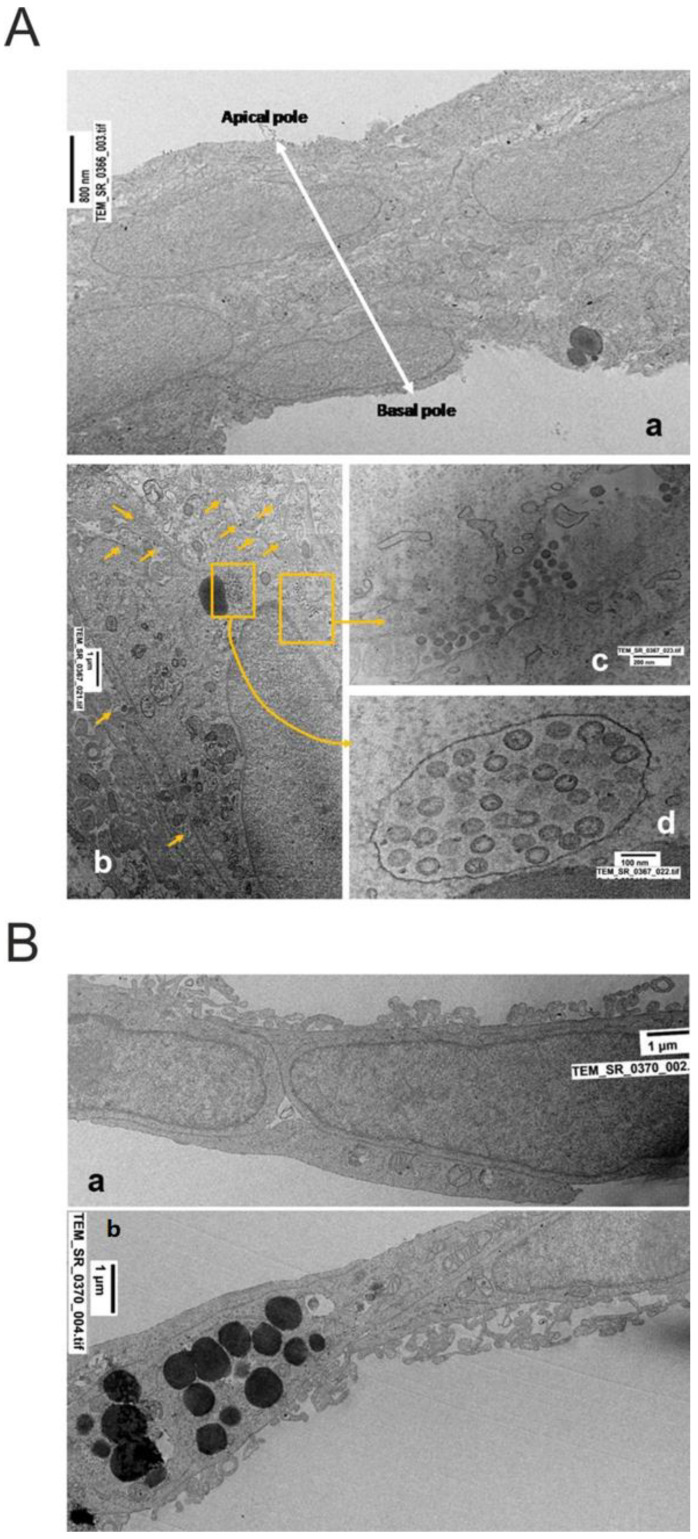
(**A**) (**a**) Control Vero E6 cells. (**b**) SARS-CoV-2 infected Vero E6 cells; (**c**) detail of virus particles in the extracellular space; (**d**) cytoplasmic vacuole containing numerous virus particles with spiky surfaces. Magnification is indicated by bars in each micrograph. *For description see text*. (**B**) (**a**,**b**) Control A549 cells. Two major subpopulations of cells were present; the first (**a**) was characterized by large nuclei and scarce cytoplasm, containing a few mitochondria and other organelles, with absent lipid bodies. The second type of cells (**b**) was larger with abundant cytoplasm containing numerous electron-dense granules (likely immature surfactant).

**Figure 3 ijms-23-09488-f003:**
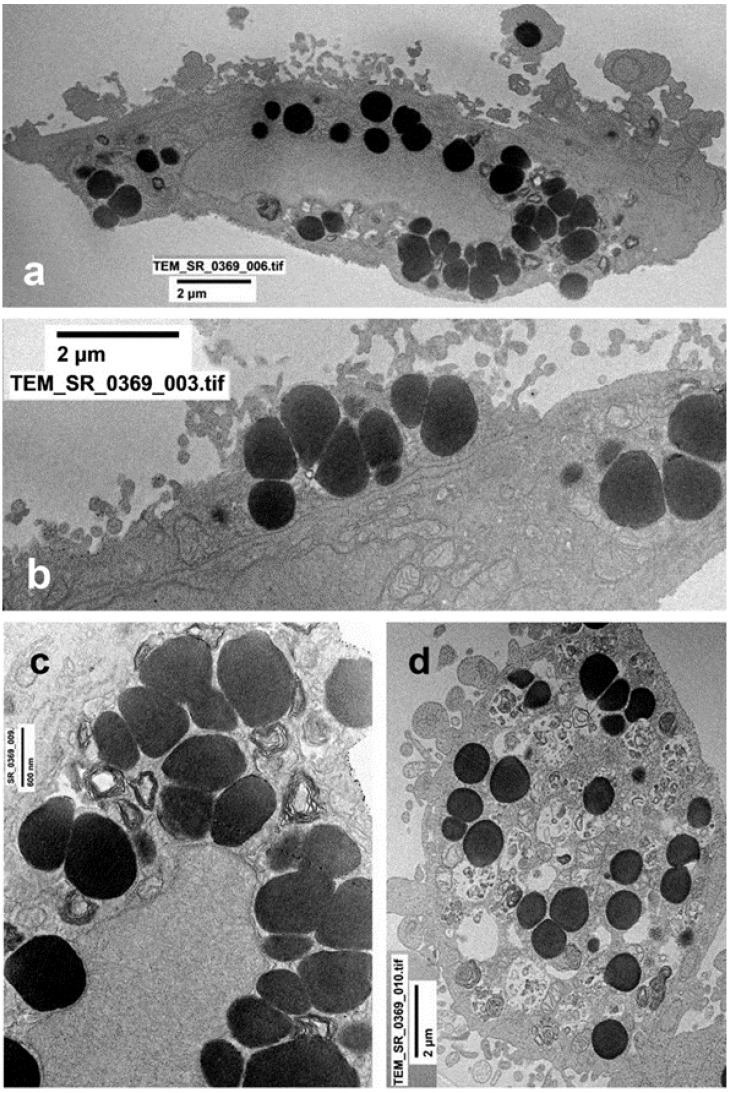
(**a**–**d**) Cells treated with nicotine alone for 24 h. (**a**) The majority of cells were larger than control, with abundant cytoplasm and numerous electron-dense granules of immature surfactant. Occasionally, granules appeared to be released from cytoplasm (arrowheads) or as microvescicles (arrow). (**b**) Detail of peripheral large dense granules during secretion. (**c**) Cytoplasm, as compared to untreated cells, contained a larger number of osmiophilic lipid bodies. (**d**) Three types of granules were observed: 1. very dense and homogeneous; 2. decreased density and homogeneity; 3. vacuoles containing myelin-like osmiophilic membranes (final differentiation).

**Figure 4 ijms-23-09488-f004:**
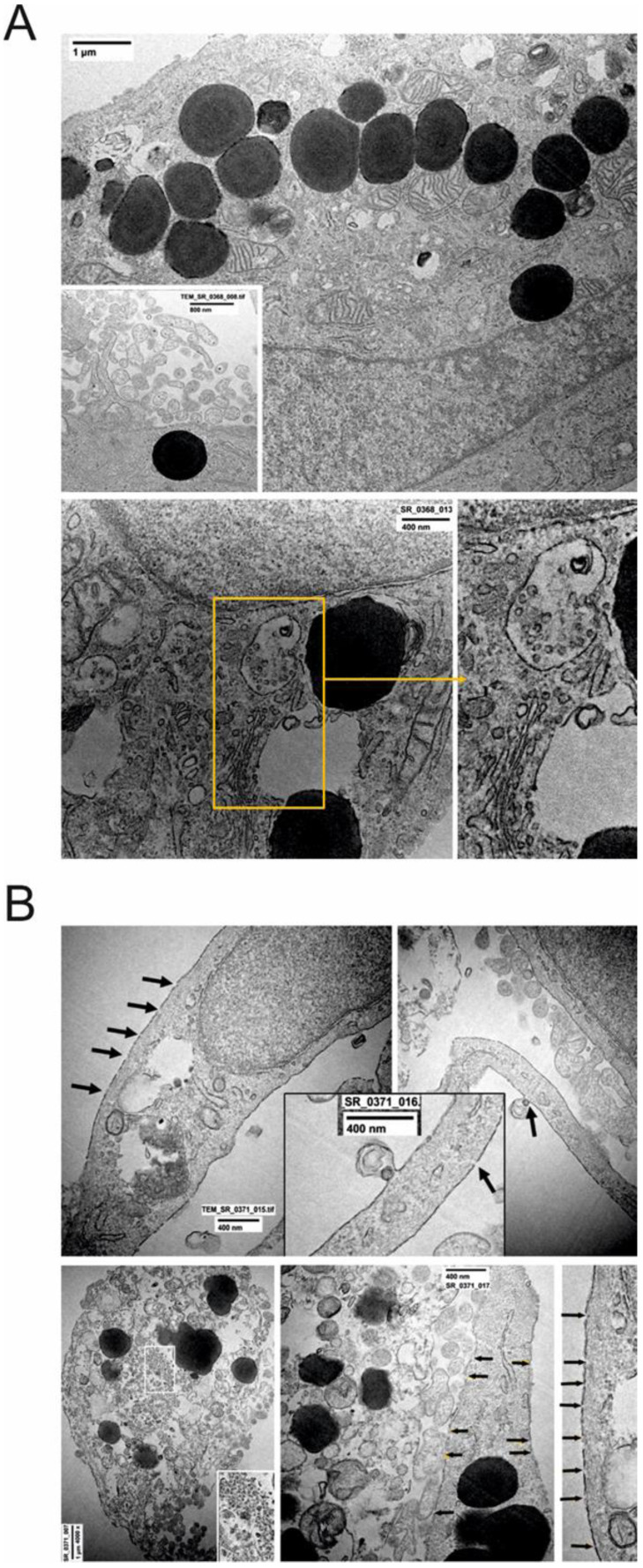
(**A**) Effects of SARS-CoV-2 treatment on A549 cell ultrastructure. (a) Well-preserved cells; a large number of plasmamembrane protrusions and vesiscles (insert). (b) Cytopathic effects; multivesicular body thought to be viral factory. Vesiculation, autophagocytosis, Golgi alterations, reticulum fragmentation, and mitochondrial swelling were present.

## Data Availability

Not applicable.

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
