# Peer review of "Nicotine in Combination with SARS-CoV-2 Affects Cells Viability, Inflammatory Response and Ultrastructural Integrity"

_ijms, 2022, doi:10.3390/ijms23169488_

Round 1

Reviewer 1 Report

Sansone L et al.’s manuscript titled “Nicotine in combination with SARS-COV-2 affects cells viability, inflammatory response, and ultrastructural integrity” investigated the effect of Nicotin and SARS-COV-2 on cell viability and inflammatory response using lung cells lines in vitro. This study showed that nicotine treatment increased the levels of SARS-CoV- 2 induced cytokines. Further analysis demonstrated that ultrastructural modification was affected by the combination treatment. In general, this work somehow looks like a very general descriptive observation. This study is straightforward but lacks the mechanism underlying the Nicotine and SARS-COV-2 combination treatment. Also, the author did not prove any novel hypothesis based on their finding.  

Reviewer 2 Report

The work of Sansone et al. is certainly interesting and innovative. Considering that too little is known about the SARS-COV-2 virus and its possible biological cross-talks, this study can provide a huge contribution, elucidating unknown mechanisms and dispelling some myths. However, I believe that some changes need to be made before acceptance, especially in the text and quality of presentation.

Comments

Lines 4-7. The "(PhD)" should be removed next to each author, it is not required by the MDPI guidelines.

Lines 54-55. This part of the introduction would seem more suitable for a review than a research article. I suggest remodeling it, adding a little more background on COVID-19 at the beginning.

Line 100. "2." should be deleted at the end of the sentence.

The Introduction should be one big paragraph, spaces should be eliminated.

In the Results the figures should be close to their citation

Lines 197-198. These phrases belong to the MDPI template, they must be deleted

Figure 1. I appreciate that the authors have reported the curve of the ELISA Kits as a miniature of the figure. This gives value to the paper.

Like for the Introduction, also the Discussion should be one big paragraph, spaces should be eliminated.

Figure 5. Figure 5 looks more like a graphical abstract and not a figure to be included in the results

Line 291, 292 and similar. In my opinion, references to figures should be removed in the Discussion. Despite this, the Discussion appears well articulated.

4. Materials and Methods. The section concerning the materials used is missing. Everything should be written in detail. Including culture media and standard growth conditions for cell cultures. In addition, the ELISA kit codes are missing.

Round 2

Reviewer 1 Report

Thanks for the revision, I have no further comments.

Reviewer 2 Report

The authors well replied to my previous comments improving the quality of the manuscript, which now is acceptable.